# Capacity of strong attractor patterns to model behavioural and cognitive prototypes

**Abbas Edalat**
Department of Computing
Imperial College London
London SW72RH, UK
ae@ic.ac.uk

## Abstract

We solve the mean field equations for a stochastic Hopfield network with temperature (noise) in the presence of strong, i.e., multiply stored, patterns, and use this solution to obtain the storage capacity of such a network. Our result provides for the first time a rigorous solution of the mean filed equations for the standard Hopfield model and is in contrast to the mathematically unjustifiable replica technique that has been used hitherto for this derivation. We show that the critical temperature for stability of a strong pattern is equal to its degree or multiplicity, when the sum of the squares of degrees of the patterns is negligible compared to the network size. In the case of a single strong pattern, when the ratio of the number of all stored pattens and the network size is a positive constant, we obtain the distribution of the overlaps of the patterns with the mean field and deduce that the storage capacity for retrieving a strong pattern exceeds that for retrieving a simple pattern by a multiplicative factor equal to the square of the degree of the strong pattern. This square law property provides justification for using strong patterns to model attachment types and behavioural prototypes in psychology and psychotherapy.

## 1 Introduction: Multiply learned patterns in Hopfield networks

The Hopfield network as a model of associative memory and unsupervised learning was introduced in [23] and has been intensively studied from a wide range of viewpoints in the past thirty years. However, properties of a strong pattern, as a pattern that has been multiply stored or learned in these networks, have only been examined very recently, a surprising delay given that repetition of an activity is the basis of learning by the Hebbian rule and long term potentiation. In particular, while the storage capacity of a Hopfield network with certain correlated patterns has been tackled [13, 25], the storage capacity of a Hopfield network in the presence of strong as well as random patterns has not been hitherto addressed.

The notion of a strong pattern of a Hopfield network has been proposed in [15] to model attachment types and behavioural prototypes in developmental psychology and psychotherapy. This suggestion has been motivated by reviewing the pioneering work of Bowlby [9] in attachment theory and highlighting how a number of academic biologists, psychiatrists, psychologists, sociologists and neuroscientists have consistently regarded Hopfield-like artificial neural networks as suitable tools to model cognitive and behavioural constructs as patterns that are deeply and repeatedly learned by individuals [11, 22, 24, 30, 29, 10].

A number of mathematical properties of strong patterns in Hopfield networks, which give rise to strong attractors, have been derived in [15]. These show in particular that strong attractors are strongly stable; a series of experiments have also been carried out which confirm the mathematical

results and also indicate that a strong pattern stored in the network can be retrieved even in the presence of a large number of simple patterns, far exceeding the well-known maximum load parameter or storage capacity of the Hopfield network with random patterns ($\alpha_c \approx 0.138$).

In this paper, we consider strong patterns in stochastic Hopfield model with temperature, which accounts for various types of noise in the network. In these networks, the updating rule is probabilistic and depend on the temperature. Since analytical solution of such a system is not possible in general, one strives to obtain the average behaviour of the network when the input to each node, the so-called field at the node, is replaced with its mean. This is the basis of mean field theory for these networks.

Due to the close connection between the Hopfield network and the Ising model in ferromagnetism [1, 8], the mean field approach for the Hopfield network and its variations has been tackled using the replica method, starting with the pioneering work of Amit, Gutfreund and Sompolinsky [3, 2, 4, 19, 31, 1, 13]. Although this method has been widely used in the theory of spin glasses in statistical physics [26, 16] its mathematical justification has proved to be elusive as we will discuss in the next section; see for example [20, page 264], [14, page 27], and [7, page 9].

In [17] and independently in [27], an alternative technique to the replica method for solving the mean field equations has been proposed which is reproduced and characterised as heuristic in [20, section 2.5] since it relies on a number of assumptions that are not later justified and uses a number of mathematical steps that are not validated.

Here, we use the basic idea of the above heuristic to develop a verifiable mathematical framework with provable results grounded on elements of probability theory, with which we assume the reader is familiar. This technique allows us to solve the mean field equations for the Hopfield network in the presence of strong patterns and use the results to study, first, the stability of these patterns in the presence of temperature (noise) and, second, the storage capacity of the network with a single strong pattern at temperature zero.

We show that the critical temperature for the stability of a strong pattern is equal to its degree (i.e., its multiplicity) when the ratio of the sum of the squares of degrees of the patterns to the network size tends to zero when the latter tends to infinity. In the case that there is only one strong pattern present with its degree small compared to the number of patterns and the latter is a fixed multiple of the number of nodes, we find the distribution of the overlap of the mean field and the patterns when the strong pattern is being retrieved. We use these distributions to prove that the storage capacity for retrieving a strong pattern exceeds that for a simple pattern by a multiplicative factor equal to the square of the degree of the strong attractor. This result matches the finding in [15] regarding the capacity of a network to recall strong patterns as mentioned above. Our results therefore show that strong patterns are robust and persistent in the network memory as attachment types and behavioural prototypes are in the human memory system.

In this paper, we will several times use Lyapunov's theorem in probability which provides a simple sufficient condition to generalise the Central Limit theorem when we deal with independent but not necessarily identically distributed random variables. We require a general form of this theorem as follows. Let $Y_n = \sum_{i=1}^{k_n} Y_{ni}$, for $n \in \mathbb{N}$, be a *triangular array of random variables* such that for each $n$, the random variables $Y_{ni}$, for $1 \leq i \leq k_n$ are independent with $\mathrm{E}(Y_{ni}) = 0$ and $\mathrm{E}(Y_{ni}^2) = \sigma_{ni}^2$, where $\mathrm{E}(X)$ stands for the expected value of the random variable $X$. Let $s_n^2 = \sum_{i=1}^{k_n} \sigma_{ni}^2$. We use the notation $X \sim Y$ when the two random variables $X$ and $Y$ have the same distribution (for large $n$ if either or both of them depend on $n$).

**Theorem 1.1** *(Lyapunov's theorem [6, page 368]) If for some $\delta > 0$, we have the condition:*

$$\frac{1}{s_n^{2+\delta}} E(|Y_n|^{2+\delta}|) \to 0 \qquad as \ n \to \infty$$

*then $\frac{1}{s_n} Y_n \xrightarrow{\mathrm{d}} \mathcal{N}(0,1)$ as $n \to \infty$ where $\xrightarrow{\mathrm{d}}$ denotes convergence in distribution, and we denote by $\mathcal{N}(a, \sigma^2)$ the normal distribution with mean $a$ and variance $\sigma^2$. Thus, for large $n$ we have $Y_n \sim \mathcal{N}(0, s_n^2)$.* $\square$

## 2  Mean field theory

We consider a Hopfield network with $N$ neurons $i = 1, \ldots, N$ with values $S_i = \pm 1$ and follow the notations in [20]. As in [15], we assume patterns can be multiply stored and the *degree* of a pattern is defined as its multiplicity. The total number of patterns, counting their multiplicity, is denoted by $p$ and we assume there are $n$ patterns $\xi^1, \ldots, \xi^n$ with degrees $d_1, \ldots, d_n \geq 1$ respectively and that the remaining $p - \sum_{k=1}^{n} d_k \geq 0$ patterns are simple, i.e., each has degree one. Note that by our assumptions there are precisely

$$p_0 = p + n - \sum_{k=1}^{n} d_k$$

distinct patterns, which we assume are independent and identically distributed with equal probability of taking value $\pm 1$ for each node. More generally, for any non-negative integer $k \in \mathbb{N}$, we let

$$p_k = \sum_{\mu=1}^{p_0} d_\mu^k.$$

We use the generalized Hebbian rule for the synaptic couplings: $w_{ij} = \frac{1}{N} \sum_{\mu=1}^{p_0} d_\mu \xi_i^\mu \xi_j^\mu$ for $i \neq j$ with $w_{ii} = 0$ for $1 \leq i, j \leq N$. As in the standard stochastic Hopfield model [20], we use Glauber dynamics [18] for the stochastic updating rule with pseudo-temperature $T > 0$, which accounts for various types of noise in the network, and assume zero bias in the local field. Putting $\beta = 1/T$ (i.e., with the Boltzmann constant $k_B = 1$) and letting $f_\beta(h) = 1/(1 + \exp(-2\beta h))$, the stochastic updating rule at time $t$ is given by:

$$\Pr(S_i(t+1) = \pm 1) = f_\beta(\pm h_i(t)), \quad \text{where } h_i(t) = \sum_{j=1}^{N} w_{ij} S_j(t), \tag{1}$$

is the local field at $i$ at time $t$. The updating is implemented asynchronously in a random way.

The energy of the network in the configuration $S = (S_i)_{i=1}^{N}$ is given by

$$H(S) = -\frac{1}{2} \sum_{i,j=1}^{N} S_i S_j w_{ij}.$$

For large $N$, this specifies a complex system, with an underlying state space of dimension $2^N$, which in general cannot be solved exactly. However, mean field theory has proved very useful in studying Hopfield networks. The average updated value of $S_i(t+1)$ in Equation (1) is

$$\langle S_i(t+1) \rangle = 1/(1 + e^{-2\beta h_i(t)}) - 1/(1 + e^{2\beta h_i(t)}) = \tanh(\beta h_i(t)), \tag{2}$$

where $\langle \ldots \rangle$ denotes taking average with respect to the probability distribution in the updating rule in Equation (1). The stationary solution for the mean field thus satisfies:

$$\langle S_i \rangle = \langle \tanh(\beta h_i) \rangle, \tag{3}$$

The average overlap of pattern $\xi^\mu$ with the mean field at the nodes of the network is given by:

$$m_\nu = \frac{1}{N} \sum_{i=1}^{N} \xi_i^\nu \langle S_i \rangle \tag{4}$$

The replica technique for solving the mean field problem, used in the case $p/N = \alpha > 0$ as $N \to \infty$, seeks to obtain the average of the overlaps in Equation (4) by evaluating the partition function of the system, namely,

$$Z = \text{Tr}_S \exp(-\beta H(S)),$$

where the trace $\text{Tr}_S$ stands for taking sum over all possible configurations $S = (S_i)_{i=1}^{N}$. As it is generally the case in statistical physics, once the partition function of the system is obtained,

all required physical quantities can in principle be computed. However, in this case, the partition function is very difficult to compute since it entails computing the average $\langle\!\langle \log Z \rangle\!\rangle$ of $\log Z$, where $\langle\!\langle \ldots \rangle\!\rangle$ indicates averaging over the random distribution of the stored patterns $\xi^\mu$. To overcome this problem, the identity

$$\log Z = \lim_{k \to 0} \frac{Z^k - 1}{k}$$

is used to reduce the problem to finding the average $\langle\!\langle Z^k \rangle\!\rangle$ of $Z^k$, which is then computed for positive integer values of $k$. For such $k$, we have:

$$Z^k = \mathrm{Tr}_{S^1}\mathrm{Tr}_{S^2}\ldots\mathrm{Tr}_{S^k}\exp(-\beta(H(S^1) + H(S^1) + \ldots + H(S^k))),$$

where for each $i = 1, \ldots, k$ the super-scripted configuration $S^i$ is a *replica* of the configuration state. In computing the trace over each replica, various parameters are obtained and the replica symmetry condition assumes that these parameters are independent of the particular replica under consideration. Apart from this assumption, there are two basic mathematical problems with the technique which makes it unjustifiable [20, page 264]. Firstly, the positive integer $k$ above is eventually treated as a real number near zero without any mathematical justification. Secondly, the order of taking limits, in particular the order of taking the two limits $k \to 0$ and $N \to \infty$, are several times interchanged again without any mathematical justification.

Here, we develop a mathematically rigorous method for solving the mean field problem, i.e., computing the average of the overlaps in Equation (4) in the case of $p/N = \alpha > 0$ as $N \to \infty$. Our method turns the basic idea of the heuristic presented in [17] and reproduced in [20] for solving the *mean field equation* into a mathematically verifiable formalism, which for the standard Hopfield network with random stored patterns gives the same result as the replica method, assuming replica symmetry. In the presence of strong patterns we obtain a set of new results as explained in the next two sections.

The mean field equation is obtained from Equation (3) by approximating the right hand side of this equation by the value of $\tanh$ at the mean field $\langle h_i \rangle = \sum_{j=1}^{N} w_{ij}\langle S_j \rangle$, ignoring the sum $\sum_{j=1}^{N} w_{ij}(S_j - \langle S_j \rangle)$ for large $N$ [17, page 32]:

$$\langle S_i \rangle = \tanh(\beta\langle h_i \rangle) = \tanh\left(\tfrac{\beta}{N}\sum_{j=1}^{N}\sum_{\mu=1}^{p_0}d_\mu\xi_i^\mu\xi_j^\mu\langle S_j \rangle\right). \tag{5}$$

Equation (5) gives the mean field equation for the Hopfield network with $n$ possible strong patterns $\xi^\mu$ $(1 \le \mu \le n)$ and $p - \sum_{\mu=1}^{n} d_\mu$ simple patterns $\xi^\mu$ with $n + 1 \le \mu \le p_0$. As in the standard Hopfield model, where all patterns are simple, we have two cases to deal with. However, we now have to account for the presence of strong attractors and our two cases will be as follows: (i) In the first case we assume $p_2 := \sum_{\mu=1}^{p_0} d_\mu^2 = o(N)$, which includes the simpler case $p_2 \ll N$ when $p_2$ is fixed and independent of $N$. (ii) In the second case we assume we have a single strong attractor with the load parameter $p/N = \alpha > 0$.

## 3 Stability of strong patterns with noise: $p_2 = o(N)$

The case of constant $p$ and $N \to \infty$ is usually referred to as $\alpha = 0$ in the standard Hopfield model. Here, we need to consider the sum of degrees of all stored patterns (and not just the number of patterns) compared to $N$. We solve the mean field equation with $T > 0$ by using a method similar in spirit to [20, page 33] for the standard Hopfield model, but in our case strong patterns induce a sequence of independent but non-identically distributed random variables in the crosstalk term, where the Central Limit Theorem cannot be used; we show however that Lyapunov's theorem (Theorem (1.1) can be invoked. In retrieving pattern $\xi^1$, we look for a solution of the mean filed equation of the form: $\langle S_i \rangle = m\xi_i^1$, where $m > 0$ is a constant. Using Equation (5) and separating the contribution of $\xi^1$ in the argument of $\tanh$, we obtain:

$$m\xi_i^1 = \tanh\left(\frac{m\beta}{N}\left(d_1\xi_i^1 + \sum_{j \ne i, \mu > 1}d_\mu\xi_i^\mu\xi_j^\mu\xi_j^1\right)\right). \tag{6}$$

For each $N$, $\mu > 1$ and $j \neq i$, let

$$Y_{N\mu j} = \frac{d_\mu}{N} \xi_i^\mu \xi_j^\mu \xi_j^1. \tag{7}$$

This gives $(p_0 - 1)(N - 1)$ independent random variables with $\mathrm{E}(Y_{N\mu j}) = 0$, $\mathrm{E}(Y_{N\mu j}^2) = d_\mu^2/N^2$, and $\mathrm{E}(|Y_{N\mu j}^3|) = d_\mu^3/N^3$. We have:

$$s_N^2 := \sum_{\mu>1, j\neq i} \mathrm{E}(Y_{N\mu j}^2) = \frac{N-1}{N^2} \sum_{\mu>1} d_\mu^2 \sim \frac{1}{N} \sum_{\mu>1} d_\mu^2. \tag{8}$$

Thus, as $N \to \infty$, we have:

$$\frac{1}{s_N^3} \sum_{\mu>1, j\neq i} \mathrm{E}(|Y_{N\mu j}^3|) \sim \frac{\sum_{\mu>1} d_\mu^3}{\sqrt{N}(\sum_{\mu>1} d_\mu^2)^{3/2}} \to 0. \tag{9}$$

as $N \to \infty$ since for positive numbers $d_\mu$ we always have $\sum_{\mu>1} d_\mu^3 < (\sum_{\mu>1} d_\mu^2)^{3/2}$. Thus the Lyapunov condition is satisfied for $\delta = 1$. By Lyapunov's theorem we deduce:

$$\frac{1}{N} \sum_{\mu>1, j\neq i} d_\mu \xi_i^\mu \xi_j^\mu \xi_j^1 \sim \mathcal{N}\left(0, \sum_{\mu>1} d_\mu^2/N\right) \tag{10}$$

Since we also have $p_2 = o(N)$, it follows that we can ignore the second term, i.e., the crosstalk term, in the argument of $\tanh$ in Equation (6) as $N \to \infty$; we thus obtain:

$$m = \tanh \beta d_1 m. \tag{11}$$

To examine the fixed points of the Equation (11), we let $d = d_1$ for convenience and put $x = \beta dm = dm/T$, so that $\tanh x = Tx/d$; see Figure 1. It follows that $T_c = d$ is the critical temperature. If $T < d$ then there is a non-zero (non-trivial) solution for $m$, whereas for $T > d$ we only have the trivial solution. For $d = 1$ our solution is that of the standard Hopfield network as in [20, page 34].

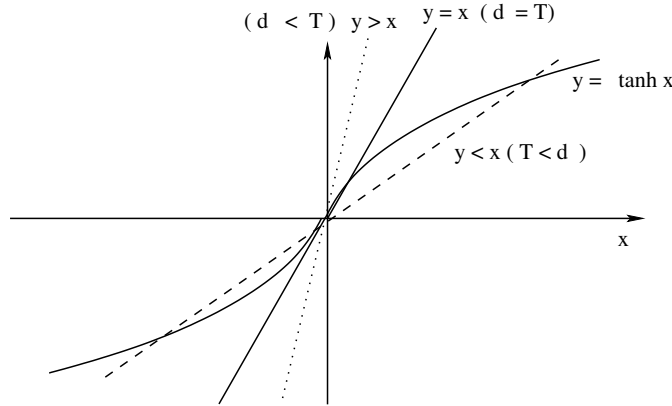

Figure 1: Stability of strong attractors with noise

**Theorem 3.1** *The critical temperature for stability of a strong attractor is equal to its degree.* $\square$

## 4 Mean field equations for $p/N = \alpha > 0$.

The case $p/N = \alpha$, as for the standard Hopfield model, is much harder and we here assume we have only a single pattern $\xi^1$ with $d_1 \geq 1$ and the rest of the patterns $\xi^\mu$ are simple with $d_\mu = 1$ for $2 \leq \mu \leq p_0$. The case when there are more than one strong patterns is harder and will be dealt with in a future paper. Moreover, we assume $d_1 \ll p_0$ which is the interesting case in applications. If $d_1 > 1$ then we have a single strong pattern whereas if $d_1 = 1$ the network is reduced to the standard Hopfield network. We recall that all patterns $\xi^\mu$ for $1 \leq \mu \leq p_0$ are independent and random. Since

$p$ and $N$ are assumed to be large and $d_1 \ll p_0$, we will replace $p_0$ with $p$ and approximate terms like $p - 2$ with $p$.

We again consider the mean field equation (5) for retrieving pattern $\xi^1$ but now the cross talk term in (6) is large and can no longer be ignored. We therefore look at the overlaps, Equation (4), of the mean field with all the stored patterns $\xi^\nu$ and not just $\xi^1$.

Combining Equation (5) and (4), we eliminate the mean field to obtain a recursive equation for the overlaps as the new variables:

$$m_\nu = \frac{1}{N} \sum_{i=1}^{N} \xi_i^\nu \tanh\left(\beta \sum_{\mu=1}^{p} d_\mu \xi_i^\mu m_\mu\right) \tag{12}$$

We now have a family of $p$ stochastic equations for the random variables $m_\nu$ with $1 \le \nu \le p$ in order to retrieve the random pattern $\xi^1$. Formally, we assume we have a probability space $(\Omega, \mathcal{F}, P)$ with the real-valued random variables $m_\nu : \Omega \to I\!R$, which are measurable with respect to $\mathcal{F}$ and the Borel sigma field $\mathcal{B}$ over the real line and which take value $m_\nu(\omega) \in I\!R$ for each sample point $\omega \in \Omega$. The probability of an event $A \in \mathcal{B}$ is given by $\Pr\{\omega : m_\nu(\omega) \in A\}$. As usual $\Omega$ can itself be taken to the real line with its Borel sigma field and we will usually drop all references to $\Omega$. We need two lemmas to prove our main result. We write $X_N \xrightarrow{\text{a.s.}} X$ for the almost sure convergence of the sequence of random variables $X_N$ to $X$, whereas $X_N \xrightarrow{\text{d}} X$ indicates convergence in distribution [6]. Recall that almost sure convergence implies convergence in distribution. To help us compute the right hand side of Equation (12), we need the following lemma, which extends the standard result for the Law of Large Numbers and its rate of convergence [5, pages 112 and 113].

**Lemma 4.1** *Let $X$ be a random variable on $I\!R$ such that its probability distribution $F(x) = Pr(X \le x)$ is differentiable with density $F'(x) = f(x)$. If $g : I\!R \to I\!R$ is a bounded measurable function and $X_k$ ($k \ge 1$) is a sequence of of independent and identically distributed random variables with distribution $X$, then*

$$\frac{1}{N} \sum_{i=1}^{N} g(X_i) \xrightarrow{\text{a.s.}} Eg(X) = \int_{\infty}^{\infty} g(x) f(x) dx, \tag{13}$$

*and for all $\epsilon > 0$ and $t > 1$, we have:*

$$\Pr\left(\sup_{k \ge N}\left(\frac{1}{k}\sum_{i=1}^{k}(g(X_i) - kE(g)(X))\right) \ge \epsilon\right) = o(1/N^{t-1}) \quad \square \tag{14}$$

The proof of the above lemma is given on-line in the supplementary material.

Assume $p/N = \alpha > 0$ with $d_1 \ll p_0$ and $d_\mu = 1$ for $1 < \mu \le p_0$. In the following theorem, we use the basic idea of the heuristic in [17] which is reproduced in [20, section 2.5] to develop a verifiable mathematical method with provable results to solve the mean field equation in the more general case that we have a single strong pattern present in the network.

**Theorem 4.2** *There is a solution to the mean field equations (12) for retrieving $\xi^1$ with independent random variables $m_\nu$ (for $1 \le \nu \le p_0$), where $m_1 \sim \mathcal{N}(m, s/N)$ and $m_\nu \sim \mathcal{N}(0, r/N)$ (for $\nu \ne 1$), if the real numbers $m$, $s$ and $r$ satisfy the four simultaneous equations:*

$$\begin{cases} \text{(i)} & m = \int_{-\infty}^{\infty} \frac{dz}{\sqrt{2\pi}} e^{-\frac{z^2}{2}} \tanh(\beta(d_1 m + \sqrt{\alpha r}z)) \\ \text{(ii)} & s = q - m^2 \\ \text{(iii)} & q = \int_{-\infty}^{\infty} \frac{dz}{\sqrt{2\pi}} e^{-\frac{z^2}{2}} \tanh^2(\beta(d_1 m + \sqrt{\alpha r}z)) \\ \text{(iv)} & r = \frac{q}{(1 - \beta(1-q))^2} \end{cases} \tag{15}$$

In the proof of this theorem, as given below, we seek a solution of the mean field equations assuming we have independent random variables $m_\nu$ (for $1 \le \nu \le p_0$) such that for large $N$ and $p$ with

$p/N = \alpha$, we have $m_1 \sim \mathcal{N}(m, s/N)$ and $m_\nu \sim \mathcal{N}(0, r/N)$ ($\nu \neq 1$), and then find conditions in terms of $m$, $s$ and $r$ to ensure that such a solution exists. These assumptions are in effect equivalent to the replica symmetry approximation [17, page 262], since they lead, as shown below, to the same solution derived from the replica method when all stored patterns are simple. In analogy with the replica technique, we call our solution *symmetric*. Since by our assumption about the distribution of the overlaps $m_\mu$, the standard deviation of each overlap is $O(1/\sqrt{N})$, we ignore terms of $O(1/N)$ and more generally terms of $o(1/\sqrt{N})$ compared to terms of $O(1/\sqrt{N})$ in the proof including in the lemma below, which enables us to compute the argument of $\tanh$ in Equation (12) for large $N$.

**Lemma 4.3** *If $m_\nu \sim \mathcal{N}(0, r/N)$ (for $\nu \neq 1$), then we have the equivalence of distributions:*

$$\sum_{\mu \neq 1, \nu} \xi_i^1 \xi_i^\mu m_\mu \sim \mathcal{N}(0, \alpha r) \sim \sum_{\mu \neq 1} \xi_i^1 \xi_i^\mu m_\mu. \ \square$$

The proofs of the above lemma and Theorem (4.2) are given on-line in the supplementary material.

We note that in the heuristic described in [20] the distributions of $m_1$ and $m_\nu$ ($\nu \neq 1$) are not eventually determined yet an initial assumption about the variance of $m_\nu$ is made. Moreover, the heuristic has no assumption on how $m_\nu$ is distributed, and no valid justification is provided for computing the double summation to obtain $m_\nu$, which is similar to the lack of justification for the interchange of limits in the replica technique mentioned in Section 2.

Comparing the equations for $m$, $q$ and $r$ in Equations (15) with those obtained by the replica method [20, pages 263-4] or the heuristic in [20, page 37], we see that $m$ has been replaced by $d_1 m$ on the right hand side of the equations for $m$ and $q$. It follows that for $d_1 = 1$, we obtain the solution for random patterns in the standard Hopfield network produced by the replica method.

We can solve the simultaneous equations in (15) for $m$, $q$ and $r$ (and then for $s$) numerically. As in [20, page 38], we examine when these equations have non-trivial solutions (i.e., $m \neq 0$) when $T \to 0$ corresponding to $\beta \to \infty$, where we also have $q \to 1$ but $C := \beta(1 - q)$ remains finite: Using the relations:

$$\begin{cases} \int_{-\infty}^{\infty} \frac{dz}{\sqrt{2\pi}} e^{-z^2/2} (1 - \tanh^2 \beta(az + b)) \approx \frac{2}{\pi} \frac{1}{a\beta} e^{-b^2/2a^2} \\ \int_{-\infty}^{\infty} \frac{dz}{\sqrt{2\pi}} e^{-z^2/2} \tanh \beta(az + b) \overset{\beta \to \infty}{\longrightarrow} \mathrm{erf}(b/\sqrt{2}a), \end{cases} \quad (16)$$

where erf is the error function, the three equations for $m$, $q$ and $r$ become:

$$\begin{cases} C := \beta(1 - q) = \sqrt{2/\pi \alpha r} \exp(-(dm)^2/2\alpha r) \\ r = 1/(1 - C)^2, \quad m = \mathrm{erf}(dm/\sqrt{2\alpha r}), \end{cases} \quad (17)$$

where we have put $d := d_1$. Let $y = dm/\sqrt{2\alpha r}$; then we obtain:

$$f_{\alpha,d}(y) := \frac{y}{d}\left(\sqrt{2\alpha} + \frac{2}{\sqrt{\pi}} e^{-y^2}\right) = \mathrm{erf}(y) \quad (18)$$

Figure 2, gives a schematic view of the solution of Equation (18). The dotted curve is the erf function on the right hand side of the equation, whereas the three solid curves correspond to the graphs of the function $f_{\alpha,d}$ on the left hand side of the equation for a given value of $d$ and three different values of $\alpha$. The heights of these graphs increase with $\alpha$.

The critical load parameter $\alpha_c(d)$ is the threshold such that for $\alpha < \alpha_c(d)$ the strong pattern with degree $d$ can be retrieved whereas for $\alpha_c(d) < \alpha$ this memory is lost. Geometrically, $\alpha_c(d)$ corresponds to the curve that is tangent, say at $y_d$, to the error function, i.e.,

$$f'_{\alpha_c(d),d}(y_d) = \mathrm{erf}'(y_d).$$

For $\alpha < \alpha_c(d)$, the function $f_{\alpha,d}$ has two non-trivial intersections (away from the origin) with erf while for $\alpha_c(d) < \alpha$ there are no non-trivial intersections.

We can compare the storage capacity of strong patterns with that of simple patterns, assuming the independence of $m_\nu$ (equivalently replica symmetry), by finding a lower bound for $\alpha_c(d)$ in terms

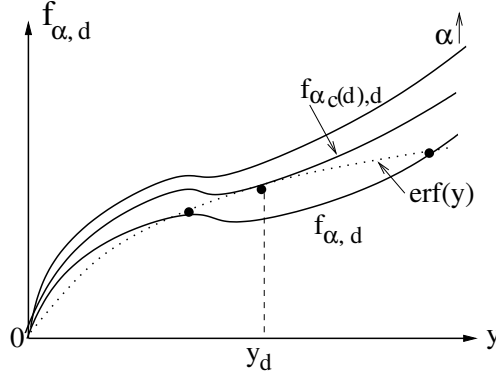

Figure 2: Capacity of strong attractors

of $\alpha_c(1)$ as follows. We have:

$$f_{\alpha,d}(y) = y(\sqrt{2(\alpha/d^2)} + \frac{2}{d\sqrt{\pi}}e^{-y^2}) \le y(\sqrt{2(\alpha/d^2)} + \frac{2}{\sqrt{\pi}}e^{-y^2}) \qquad (19)$$

where equality holds iff $d = 1$. Putting $\alpha = d^2\alpha_c(1)$ and $y = y_1$, we have for $d > 1$:

$$f_{d^2\alpha_c(1),d}(y_1) < f_{\alpha_c(1),1}(y_1) = \text{erf}(y_1), \qquad (20)$$

Therefore, for a strong pattern, the graphs of $f_{d^2\alpha_c(1),d}$ and erf intersect in two non-trivial points and thus $\alpha_c(d) > d^2\alpha_c(1)$. Since $\alpha_c(1) = \alpha_c \approx 0.138$, this yields: $\alpha_c(d)/0.138 > d^2$, i.e., the relative increase in the storage capacity exceeds the square of the degree of the strong pattern.

In the case of the standard Hopfield network with simple patterns only, we have $\alpha_c(1) = \alpha_c \approx 0.138$, but simulation experiments show that for values in the narrow range $0.138 < \alpha < 0.144$ there are replica symmetry breaking solutions for which a stored pattern can still be retrieved [12]. We show that the square property holds when we take into account symmetry breaking solutions. By [15, Theorem 1], it follows that the error probability of retrieving a single strong attractor is:

$$\text{Pr}_{er} \approx \frac{1}{2}(1 - \text{erf}(d/\sqrt{2\alpha}),$$

for $\alpha = p/N$. Thus, this error will be constant if $d/\sqrt{\alpha}$ remains fixed, indicating that the critical value of the load parameter is proportional to the square of the degree of the strong attractor.

**Corollary 4.4** *The storage capacity for retrieving a single strong pattern exceeds that of a simple pattern by the square of the degree of the strong pattern.* $\square$

This square property shows that a multiply learned pattern is retained in the memory in the presence of a large number of other random patterns, proportional to the square of its multiplicity.

## 5   Conclusion

We have developed a mathematically justifiable method to derive the storage capacity of the Hopfield network when the load parameter $\alpha = p/N$ remains a positive constant as the network size $N \to \infty$. For the standard model, our result confirms that of the replica technique, i.e., $\alpha_c \approx 0.138$. However, our method also computes the storage capacity when retrieving a single strong pattern of degree $d$ in the presence of other random patterns and we have shown that this capacity exceeds that of a simple pattern by a multiplicative factor $d^2$, providing further justification for using strong patterns of Hopfield networks to model attachment types and behavioural prototypes in psychology.

The storage capacity of Hopfield networks when there are more than a single strong pattern and in networks with low neural activation will be addressed in future work. It is also of interest to examine the behaviour of strong patterns in Boltzmann Machines [20], Restricted Boltzmann Machines [28] and Deep Learning Networks [21].

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
