[Supplementary Material]

# Supplemtary Material:
# Capacity of strong attractor patterns to model behavioural and cognitive prototypes

**Abbas Edalat**
Department of Computing
Imperial College London
London SW72RH, UK
ae@ic.ac.uk

We will present the proofs of Lemma 4.1, Lemma 4.3 and Theorem 4.2 here. For completeness, first recall Lyapunov's theorem in probability theory.

Let $Y_n = \sum_{i=1}^{k_n} Y_{ni}$, for $n \in I\!N$, be a *triangular array of random variables* such that for each $n$, the random variables $Y_{ni}$, for $1 \leq i \leq k_n$ are independent with $\mathrm{E}(Y_{ni}) = 0$ and $\mathrm{E}(Y_{ni}^2) = \sigma_{ni}^2$, where $\mathrm{E}(X)$ stands for the expected value of the random variable $X$. Let $s_n^2 = \sum_{i=1}^{k_n} \sigma_{ni}^2$. We use the notation $X \sim Y$ when the two random variables $X$ and $Y$ have the same distribution (for large $n$ if either or both of them depend on $n$).

**Theorem** (Lyapunov) [2, page 368] *If for some $\delta > 0$, we have*

$$\frac{1}{s_n^{2+\delta}} E(|Y_n|^{2+\delta}|) \to 0 \qquad as \ n \to \infty$$

*then $\frac{1}{s_n} Y_n \xrightarrow{\mathrm{d}} \mathcal{N}(0,1)$ as $n \to \infty$ where $\xrightarrow{\mathrm{d}}$ denotes convergence in distribution, and we denote by $\mathcal{N}(a, \sigma^2)$ the normal distribution with mean $a$ and variance $\sigma^2$. Thus, for large $n$ we have $Y_n \sim \mathcal{N}(0, s_n^2)$.* □

**Lemma 4.1** *Let $X$ be a random variable on $I\!R$ such that its probability distribution $F(x) = Pr(X \leq x)$ is differentiable with density $F'(x) = f(x)$. If $g : I\!R \to I\!R$ is a bounded measurable function and $X_k$ $(k \geq 1)$ is a sequence of of independent and identically distributed random variables with distribution $X$, then*

$$\frac{1}{N} \sum_{i=1}^{N} g(X_i) \xrightarrow{\text{a.s.}} Eg(X) = \int_{\infty}^{\infty} g(x)f(x)dx, \tag{1}$$

*and for all $\epsilon > 0$ and $t > 1$, we have:*

$$\Pr\left(\sup_{k \geq N}\left(\frac{1}{k}\sum_{i=1}^{k}(g(X_i) - k\mathrm{E}(g)(X))\right) \geq \epsilon\right) = o(1/N^{t-1}) \tag{2}$$

**Proof** Since $g$ is bounded, $Eg(X) = \int_{\infty}^{\infty} g(x)f(x)dx$ is absolutely convergent and thus the expected value $Eg(X)$ is well-defined and $|Eg(X)| < \infty$. Equation (1) then follows from the Strong Law of Large Numbers [2, page 80] applied to the random variables $g(X_i)$, for $i \geq 1$. which are independent and identically distributed as $g(X)$ with expectation $Eg(X)$. We also have $\mathrm{E}|g(X)|^t = \int_{-\infty}^{\infty}|g(x)|^t f(x)dx < \infty$ for all $t > 1$ and thus the convergence rate of the Strong Law of Large Numbers implies Equation (2), a consequence of Theorem 3 and the lemma in [1, pages 112 and 113]. □

Assume $p/N = \alpha > 0$ with $d_1 \ll p_0$ and $d_\mu = 1$ for $1 < \mu \le p_0$. Consider the overlaps

$$m_\nu = \frac{1}{N} \sum_{i=1}^{N} \xi_i^\nu \langle S_i \rangle \tag{3}$$

and the mean field equations:

$$m_\nu = \frac{1}{N} \sum_{i=1}^{N} \xi_i^\nu \tanh\left( \beta \sum_{\mu=1}^{p} d_\mu \xi_i^\mu m_\mu \right) \tag{4}$$

**Theorem 4.2** *There is a solution to the mean field equations (4) for retrieving $\xi^1$ with independent random variables $m_\nu$ (for $1 \le \nu \le p_0$), where $m_1 \sim \mathcal{N}(m, s/N)$ and $m_\nu \sim \mathcal{N}(0, r/N)$ (for $\nu \ne 1$), if the real numbers $m$, $s$ and $r$ satisfy the four simultaneous equations:*

$$\begin{cases}
\text{(i)} & m = \int_{-\infty}^{\infty} \frac{dz}{\sqrt{2\pi}} e^{-\frac{z^2}{2}} \tanh(\beta(d_1 m + \sqrt{\alpha r} z)) \\
\text{(ii)} & s = q - m^2 \\
\text{(iii)} & q = \int_{-\infty}^{\infty} \frac{dz}{\sqrt{2\pi}} e^{-\frac{z^2}{2}} \tanh^2(\beta(d_1 m + \sqrt{\alpha r} z)) \\
\text{(iv)} & r = \frac{q}{(1 - \beta(1-q))^2}
\end{cases} \tag{5}$$

In the proof of this theorem, as given below, we seek a solution of the mean field equations assuming we have independent random variables $m_\nu$ (for $1 \le \nu \le p_0$) such that for large $N$ and $p$ with $p/N = \alpha$, we have $m_1 \sim \mathcal{N}(m, s/N)$ and $m_\nu \sim \mathcal{N}(0, r/N)$ ($\nu \ne 1$), and then find conditions in terms of $m$, $s$ and $r$ to ensure that such a solution exists. Since by our assumption about the distribution of the overlaps $m_\mu$, the standard deviation of each overlap is $O(1/\sqrt{N})$, we ignore terms of $O(1/N)$ and more generally terms of $o(1/\sqrt{N})$ compared to terms of $O(1/\sqrt{N})$ in the proof including in the lemma below.

**Lemma 4.3** If $m_\nu \sim \mathcal{N}(0, r/N)$ (for $\nu \ne 1$), then we have the equivalence of distributions:

$$\sum_{\mu \ne 1, \nu} \xi_i^1 \xi_i^\mu m_\mu \sim \mathcal{N}(0, \alpha r) \sim \sum_{\mu \ne 1} \xi_i^1 \xi_i^\mu m_\mu.$$

**Proof** Recall that the sum $\sum_{t=1}^{k} X_t$ of $k$ independent random variables such that $X_t$ has a normal distribution with mean $a_t$ and variance $\sigma_t^2$ (for $1 \le t \le k$) is itself normally distributed with mean $\sum_{t=1}^{k} a_t$ and variance $\sum_{t=1}^{k} \sigma_t^2$. Consider the first equivalence. From $-1 \le \langle S_i \rangle| \le 1$, for $1 \le i \le N$, and Equation (3), it follows that

$$\mathrm{E}\left(m_\mu \xi_j^\mu\right) = \mathrm{E}\left( \frac{1}{N} \sum_{i=1}^{N} \xi_i^\mu \langle S_i \rangle \xi_j^\mu \right) \le \mathrm{E}\left( \frac{1}{N} \sum_{i=1}^{N} \xi_i^\mu \xi_j^\mu \right) = \frac{1}{N}$$

Similarly, $\mathrm{E}(m_\mu \xi_j^\mu) \ge -1/N$, and thus $\mathrm{E}(m_\mu \xi_j^\mu) = O(1/N)$. Therefore, for $\mu \ne 1, \nu$, the three random variables $\xi_i^1$, $\xi_i^\mu$ and $m_\mu$ can be considered independent and it follows that the distribution of each product on the left hand side of the first equivalence is given by $\mathcal{N}(0, r/N)$. Summing up the approximately $p$ independent normal distributions $\mu \ne 1, \nu$, we obtain the first equivalence. The second equivalence is proved in a similar way. $\square$

**Proof of Theorem 4.2** First consider Equation (4) for $\nu = 1$, which, by separating the contributions of $\mu = 1$ and $\mu \ne 1$ on the right hand side, we write as

$$m_1 = Y_N := \frac{1}{N} \sum_{i=1}^{N} \xi_i^1 \tanh \beta(d_1 m_1 \xi_i^1 + \sum_{\mu \ne 1} \xi_i^\mu m_\mu). \tag{6}$$

Multiplying the odd function $\tanh$ and its argument by $\xi_i^1$, we obtain:

$$\begin{cases}
Y_N = \frac{1}{N} \sum_{i=1}^{N} \xi_i^1 \xi_i^1 \tanh \beta(d_1 m_1 \xi_i^1 \xi_i^1 + \sum_{\mu \ne 1} \xi_i^\mu \xi_i^1 m_\mu) \\
= \frac{1}{N} \sum_{i=1}^{N} \tanh \beta(d_1 m_1 + \sum_{\mu \ne 1} \xi_i^\mu \xi_i^1 m_\mu) \\
\xrightarrow{\text{a.s.}} \int_{-\infty}^{\infty} \frac{dz}{\sqrt{2\pi}} e^{-\frac{z^2}{2}} \tanh(\beta(m d_1 + \sqrt{\alpha r} z)),
\end{cases} \tag{7}$$

where the last step is justified as follows. By Lemma 4.3

$$\sum_{\mu \neq 1} \xi_i^\mu \xi_i^1 m_\mu \sim \mathcal{N}(0, \alpha r) \tag{8}$$

Since, by assumption, $m_1$ has distribution $\mathcal{N}(m, s/N)$ and is independent of $\sum_{\mu \neq 1} \xi_i^\mu \xi_i^1 m_\mu$, it follows that $d_1 m_1 + \sum_{\mu \neq 1} \xi_i^\mu \xi_i^1 m_\mu$ is the sum of two normal distributions and thus has itself normal distribution $\mathcal{N}(d_1 m, \frac{d_1^2 s}{N} + r\alpha) \sim \mathcal{N}(d_1 m, r\alpha)$ by ignoring $\frac{d_1^2 s}{N}$ compared to $r\alpha$:

$$X_i := d_1 m_1 + \sum_{\mu \neq 1} \xi_i^\mu \xi_i^1 m_\mu \sim \mathcal{N}(d_1 m, r\alpha) \tag{9}$$

Therefore, the random variables $X_i$, for $i \geq 1$, are independent and identically distributed with distribution $\sim \mathcal{N}(d_1 m, \alpha r)$, and the last step in Equation (7) then follows by applying Lemma 4.1 using $g(x) = \tanh(\beta x)$, which is a bounded continuous function. Since almost sure convergence implies convergence in distribution, it follows that as $N \to \infty$,

$$Y_N \xrightarrow{\text{d}} \int_{-\infty}^{\infty} \frac{dz}{\sqrt{2\pi}} e^{-\frac{z^2}{2}} \tanh(\beta(d_1 m + \sqrt{\alpha r} z)), \tag{10}$$

where the latter is the degenerate (point) distribution with the integral on the right hand side as its value. On the other hand, by the assumption about $m_1$, we have

$$m_1 \sim \mathcal{N}(m, s/N) \xrightarrow{\text{d}} m, \tag{11}$$

as $N \to \infty$. Therefore, from Equations (6), (11) and (10), we can now obtain

$$m = \int_{-\infty}^{\infty} \frac{dz}{\sqrt{2\pi}} e^{-\frac{z^2}{2}} \tanh(\beta(d_1 m + \sqrt{\alpha r} z)), \tag{12}$$

which gives Equation (5(i)).

Next, write $Y_N = \sum_{i=1}^{N} Y_{Ni}$ with $Y_{Ni} = \frac{1}{N} \tanh \beta(X_i)$. We have a triangular array of random variables with $\text{E}(Y_{Ni}) = m/N$, by Equation (9), the equality in Equation (1), using $g(x) = \frac{1}{N} \tanh \beta(x)$ and $f$ as the Gaussian distribution $\mathcal{N}(d_1 m, r\alpha)$, and Equation (12). Furthermore,

$$\text{E}(Y_{Ni}^2) = q/N^2, \tag{13}$$

by Equation (9), where $q$ is given in Equation (5(iii)). This gives

$$\sigma_{Ni}^2 := \text{E}(Y_{Ni}^2) - (\text{E}(Y_{Ni}))^2 = (q - m^2)/N^2, \qquad \sigma_N^2 := \sum_{i=1}^{N} \sigma_{Ni}^2 = (q - m^2)/N.$$

Moreover, it is easy to see that $\text{E}(|Y_{Ni}|^3) \leq 1/N^3$ since $\tanh$ is bounded by 1. Thus,

$$\frac{1}{\sigma_N^3} \sum_{i=1}^{N} \text{E}(|Y_{Ni}|^3) = O(1/N^{1/2}) \tag{14}$$

and it follows that the Lyapunov condition holds for $\delta = 1$. Therefore, by Lyapunov's theorem $(Y_N - m)/\sigma_N \sim \mathcal{N}(0, 1)$, as $N \to \infty$, and thus $m_1 = Y_N \sim \mathcal{N}(m, (q - m^2)/N)$, as $N \to \infty$. Since by assumption $m_1 \sim \mathcal{N}(m, s/N)$, we obtain the value of $s = q - m^2$ as in Equation (5(ii)).

Now fix $\nu \neq 1$ in Equation (4), take a sample point $\omega \in \Omega$, separate the three terms for $\mu = 1$, $\mu = \nu$ and $\mu \neq 1, \nu$ on the right hand side of the equation, as before multiply $\tanh$ and its argument by $\xi_i^1$ and write the equation as $m_\nu(\omega) = h(m_\nu(\omega))$, where $h : \mathbb{R} \to \mathbb{R}$ with

$$h(x) = \frac{1}{N} \sum_{i=1}^{N} \xi_i^\nu(\omega) \xi_i^1(\omega) \tanh \beta \left( d_1 m_1(\omega) + \xi_i^\nu(\omega)\xi_i^1(\omega)x + \sum_{\mu \neq 1, \nu} \xi_i^\nu(\omega)\xi_i^1(\omega) m_\mu(\omega) \right) \tag{15}$$

By assumption $m_\nu$ is normally distributed with mean zero and standard deviation $\sqrt{r}/\sqrt{N}$. Therefore, in contrast to the case for $m_1$ treated earlier, here $m_\nu(\omega)$ is small and of order $O(1/\sqrt{N})$. Since $m_\nu$ appears in two terms on both sides of $m_\nu(\omega) = h(m_\nu(\omega))$, we need to collect together on one side of the equation the contributions of these two terms. To this end, we regard $m_\nu(\omega)$ as small compared with the term $m_1(\omega)d_1$ and the term $\sum_{\mu \neq 1,\nu} \xi_i^1(\omega)\xi_i^\mu(\omega)m_\mu(\omega)$ which are both of order $O(1)$, and we employ the Taylor expansion of $h$ near the origin $x = 0$:

$$
\begin{aligned}
h(m_\nu(\omega)) &= \tfrac{1}{N}\sum_{i=1}^N \xi_i^\nu(\omega)\xi_i^1(\omega)\tanh\beta(d_1 m_1(\omega) + \sum_{\mu\neq 1,\nu}\xi_i^\mu(\omega)\xi_i^1(\omega)m_\mu(\omega)) \\
&+ \tfrac{\beta}{N}\left(\sum_{i=1}^N(1-\tanh^2(\beta(d_1 m_1(\omega) + \sum_{\mu\neq 1,\nu}\xi_i^\mu(\omega)\xi_i^1(\omega)m_\mu(\omega))))\right)m_\nu(\omega) + c(m_\nu(\omega))^2
\end{aligned}
$$

(16)

where $|c| \leq \beta^2$, which is obtained by using the Lagrange form of remainder $c(m_\nu(\omega))^2$ to estimate the second derivative $h''(0)$ and by noting that $|\tanh(x)| \leq 1$ for all $x \in \mathbb{R}$. Thus, the Taylor series remainder is of order $O(1/N)$, which we ignore compared to the standard deviation of $m_\nu$ namely $\sqrt{r}/\sqrt{N}$. By Lemma 4.1, the last summation in Equation (16), containing the bounded continuous function $\tanh^2$, converges almost surely to $q$ as $N \to \infty$. Moreover, by using $t = 3/2$ in the second part of Lemma 4.1, it follows that for large $N$, while retaining $m_\nu$ which is of order $1/\sqrt{N}$, we can replace the sum in the equation with $q$ by ignoring the error which, for any degree of certainty, is of order $o(1/\sqrt{N})$. Thus, by using $m_\nu(\omega) = h(m_\nu(\omega))$ from Equation (4), we now obtain the following reduced stochastic equation for $\nu \neq 1$:

$$
(1 - \beta(1-q))m_\nu(\omega) = \frac{1}{N}\sum_{i=1}^N \xi_i^\nu(\omega)\xi_i^1(\omega)\tanh\beta\left(d_1 m_1(\omega) + \sum_{\mu\neq 1,\nu}\xi_i^\mu(\omega)\xi_i^1(\omega)m_\mu(\omega)\right)
$$

(17)

Now we drop $\omega$ everywhere and let the right hand side of Equation (17) be written as $Z_N = \sum_{i=1}^N Z_{Ni}$ with $Z_{Ni} = \frac{1}{N}\xi_i^\nu\xi_i^1\tanh\beta(X_i')$, where $X_i' = d_1 m_1 + \sum_{\mu\neq 1,\nu}\xi_i^\mu\xi_i^1 m_\mu$. By Lemma 4.3 and Equation (9), $X_i' \sim X_i \sim \mathcal{N}(d_1 m, r\alpha)$ and the three random variables $\xi_i^\nu$, $\xi_i^1$ and $X_i'$ are independent.

We again have an array $Z_{Ni}$ of random variables $1 \leq i \leq N$ for each $N$, and by the independence of the above three random variables we have: $E(Z_{Ni}) = 0$ and

$$
\begin{aligned}
E(Z_{Ni}^2) &= \tfrac{1}{N^2}E(\xi_i^\nu)^2 E(\xi_i^1)^2 E(\tanh^2\beta(X_i')) \\
&= \tfrac{1}{N^2}E(\tanh^2\beta(X_i)) = E(Y_{Ni}^2) = \tfrac{q}{N^2},
\end{aligned}
$$

(18)

as in Equation (13). Therefore, $\sigma_N^2 = \sum_{i=1}^N E(Z_{Ni}^2) = q/N$. Moreover, it is easy to see that $E(|Z_{Ni}|^3) \leq 1/N^3$ since $|\tanh(x)|$ is bounded by 1 for all $x \in \mathbb{R}$. Thus,

$$
\frac{1}{\sigma_N^3}\sum_{i=1}^N E(|Z_{Ni}|^3) = O(1/N^{1/2})
$$

(19)

and it follows that the Lyapunov condition holds for $\delta = 1$. We conclude by Lyapunov's theorem that $Z_N/\sigma_N \sim \mathcal{N}(0,1)$ and thus $Z_N \sim \mathcal{N}(0, q/N)$. From this and Equation (17), we deduce that

$$
m_\nu \sim \mathcal{N}\left(0, \frac{q}{(1-\beta(1-q))^2 N}\right)
$$

(20)

and obtain $r = q/(1-\beta(1-q))^2$ in Equation (5(iv)). This completes the proof of the theorem. $\square$