[Reviews · NeurIPS 2013]

Submitted by Assigned_Reviewer_4

In this paper the authors consider applying a mean field approach to a Hopfield model which has presented with strong patterns (patterns that have been presented to the model more than once). By using a mean field approach and applying the Lyapunov condition they show that they can arrive at mean field equations for two cases:

1) When the model is presented with a finite collection of strong patterns and a collection of simple patterns such that the total number of stored patterns over the total number of neurons tends to zero as the number of neurons becomes large.

2) When the model is presented with a single strong pattern and a collection of simple patterns such that the total number of stored patterns over the total number of neurons tends to some constant alpha as the number of neurons becomes large.

In both cases they arrive at self consistent equations, in the first case they then evaluate how noise level changes the models ability to recall the strong patterns and in the second they look at how the constant alpha changes the recovery of the strong patterns.

Quality
------------------------------------------------------------
There a few typos and bad sentences in this paper (see below) but on the whole the quality of the writing is OK. My main problems, however, are with the proof/statement of lemma 4.1 and the use of lemma 4.1 in the proof of theorem 4.3.


Proof/Statement of lemma 4.1
---------------------------------
I think that lemma 4.1 should have m_mu instead of m_nu for it to be applied later to theorem 4.3 and I'm going to assume this in what follows. I think the proof of lemma 4.1 does not work. In this proof the authors assume that \xi^1, \xi_i^\mu and m_\mu are independent, this is not true since m_{\mu} is constructed from a sum including \xi_i^\mu, granted this is a dependency that becomes weaker with increasing N, but it is still present and should be addressed. Secondly the authors state that m_\nu are independent, I think this is an assumption rather than a statement since eqn 11 shows that one may be written as a function of the other m_\mu.

Proof of Theorem 4.3
--------------------------------------
I think that eqn 15 should have as the sum index mu\neq 1 and not mu\neq 1,\nu. In which case either lemma 4.1 cannot be applied at eqn 16 or the statement of lemma 4.1 must be changed. If this can be addressed then I agree that the rest of the proof of 4.3 follows.

Typos etc
--------------------------
- Index should be j around eqns 2 and 3
- I would like the authors to mention for some constant m or similar after the first eqn in section 3
- Statement of lemma 4.1 (see above)
- square at end of eqn 13 should be at the end of the proof
- brackets around the expectation in eqn 13
- lower limits of the integrals in eqn 12 and the text after eqn 13
- The paragraph above theorem 4.3 is missing a word or two I think
- Eqn 15 (see theorem 4.3 above)
- The + symbol in the variance just above and in eqn 18.

Clarity
---------------------------------------
Mistakes, typos, lemma 4.1 and the beginning of the proof of theorem 4.3 aside the paper is clear and well written. I think that if the problems outlined above were addressed the paper would present a concise and comprehensible discussion on the storage of strong patterns in the Hopfield model.

Originality
---------------------------------------
As far as I know this material is original.

Significance
----------------------------------------------
The results seem significant enough to be published.



Summary: The paper discusses a mean field approach to the Hopfield model with strong patterns. I think that there are some problems with the proofs in this paper that need to be addressed before publication in particular in lemma 4.1 and theorem 4.3. I think these problems potentially could be addressed with some tweaks here and there and a few clarifications rather than any major alterations.

After reading the author's reply to these problems I'm confident the final paper will be of a good standard. On this basis I propose it is accepted into NIPS.

Submitted by Assigned_Reviewer_5

This work is a clear extension of the work on traditional Hopfield model. In the analysis of the traditional Hopfield model, each random pattern are stored once. However, the author(s) consider a case that some patterns are learned more than one time. Those states are learned more than one times are called strong attractors. The author(s) have shown analytically that, the capacity of strong attractors can be derived as shown in paper. Also, the result for single strong attractors is claimed to be consistent with the simulation result reported in [15].

This paper is a piece of good analytical work on a simple model. This work enrich our knowledge about multiply learned patterns in Hopfield models. The author(s) have also shown the analysis clearly. To my knowledge, this work should be novel.

I think this work is a significant work. Because it supples relations about the capacity of repeatedly learned patterns in a Hopfield model. It should be of interested of computer scientists and neuroscientists. I would like to recommend to accept this paper.
Summary: This work is a nice work to show how repeatedly learning can strengthen attractor. This work should be important to other experts in the community.

Submitted by Assigned_Reviewer_6

The paper addresses a specific mathematical problem: solution of the mean field equations for a stochastic Hopfield network, under temperature (noise), in the presence of strong patterns. The authors' approach is to provide a justifiable mathematical framework for an existing heuristic solution in the literature that lacks mathematical validation. They found the critical temperature for stability of a strong pattern under certain conditions. They also found the ratio of the storage capacity for retrieving a single strong pattern vs a simple pattern exceeds the square of the degree of the strong pattern.

Quality:

As far as I can see, the authors proved the existence of the solution and gave mathematical validation of the existing heuristic. I am not an expert on the question posed in this paper, but their analysis looks sound to me. The paper looks incomplete in its current form. It seems the alternative replica method mentioned in their abstract and introduction is actually not presented. Conclusion and discussion sections are missing.

Clarity:

The paper is clearly written regarding the analysis. Nonetheless I find the introduction not clear on the broader picture of the problem that suits a general NIPS audience (see comment on Significance).

Originality:

This paper is original in terms of it is the first to provide a formal, mathematical framework for an existing, unjustified heuristic solution.

Significance:

I think this is a good paper that solves an important problem. But I think the question posed in this paper is not very well-known to a general NIPS audience, thus its background and importance need to be better described in the introduction. It also looks to me the paper, in its current form, would better suit a more specific audience of neural computing.


Summary: Solid paper; solves very specific problem that will suit a specialized audience.
Author Feedback

Author rebuttal: 1. Reviewer_4 has correctly pointed out a few simple typos in the paper (in particular m_\nu instead of m_\mu in lemma 4.1 and the summation index \mu\neq 1,\nu instead of \mu\neq 1 in the proof of theorem 4.3).

Reviewer_4 has also expressed three concerns (two in the proof of lemma 4.1 and one in the proof of theorem 4.3) and correctly states that they should be addressed. But as the reviewer has acknowledged these are not major issues at all. In fact, they are minor points which are clarified as follows:

(i) The assumption that m_\mu’s are independent should be explicitly stated which then leads to a self-consistent result.

(ii) It should be stated that for large N and fixed i the two random variables

m_\mu and

\xi_i^\mu \xi_i^1

can be considered independent since

E(m_\mu \xi_i^\mu \xi_i^1)= E(m_\mu )E(\xi_i^\mu \xi_i^1)+O(1/N)

and we can drop all terms of O(1/N) as explained in the paper. We note in passing here that in this exact setting the respectable textbook Introduction to the Theory of Neural Computation by John Hertz et al. simply says that these two random variables are independent (see page 37 line 9 of this book), which confirms that this is a minor point.

(ii) Finally, the application of lemma 4.1 to equation 15 can be made more precise by adding the case for the distribution of

\sum_ {\mu\neq 1} \xi_i^1\xi_i^\mu m_\mu

to the statement of lemma 4.1 so as to state that both of these random variables (which differ by a “negligible” random variable) have the same normal distribution for large N.

2. In response to Reviewer_6’s concerns about the introduction and conclusion, we should say that we aimed to provide, for the first time, a justifiable mathematical method to to compute the storage capacity of Hopfield networks (which works in the presence of strong patterns as well) and as a result we had no more space due to the page limitation to highlight further the significance of our results for the wider community and discuss the achievements of the paper in a concluding section.

However, we now trust that the reviewers are satisfied with the rigour and the correctness of the methodology and the proofs. Therefore, we can drop the proofs of lemmas 4.1 and 4.2 and make the proof of theorem 4.3 more concise so as to have space to expand the introduction to highlight the above points (explained in detail in section 3 below) and add a few words about the replica technique, and include a concluding section.

3. The main point in our response takes issue with the quality score and impact score given by the three reviewers. Here, we would like to raise three reasons why the paper is definitely not incremental and why it should be far more highly evaluated in terms of both quality score and impact score:

(i) In the past three decades, a score of papers, some by leading pioneers of work on neural networks such as Amit, Gutfreund and Sompolinky, have been published in leading journals, which use the unjustifiable replica method to solve mean field equations for a many variations of Hopfield networks. There are also several books on neural networks including the one by Hertz et al. cited above and one by Amit, Modeling Brain Function: The World of Attractor Neural Networks, which have used the replica method or the equally unjustifiable heuristic method to solve the mean field equations. These books remain popular and are still used by advanced undergraduate and graduate students and researchers for whom the Hopfield model remains the most basic neural model for associative memory.

Given these facts, we think an objective evaluation of a paper which for the first time in three decades provides a new and mathematically justifiable methodology for computing the storage capacity of this important class of neural networks would place it much higher.

(ii) The replica method only deals with networks with uncorrelated random patterns, whereas the methodology we have provided gives a solution also in the presence of correlated patterns, in this case a strong pattern in the presence of simple random patterns. To this effect, Lyapunov’s theorem for independent random variables that are not identically distributed is repeatedly employed in our method. This application of Lyapunov’s theorem to obtain the asymptotic behaviour of the sum of random variables, to our knowledge, is new in the context of neural networks. Thus the potential for the application of our methodology goes beyond what the existing replica technique is used for.

(iii) In our paper, we have discovered a square property for the capacity of strong patterns and thus for modelling behavioural prototypes in this way: the storage capacity of a strong pattern exceeds that of a simple pattern by a multiplicative factor equal to the square of the degree of the strong pattern. As we have briefly mentioned in the Introduction, this property explains why attachment types, cognitive and behavioural prototypes are so addictive: If in a network with N neurons, a strong pattern is learned d times then even if
d^2 x 0.138 x N simple patterns are learned, we will still retrieve the strong pattern with very high probability whenever the network is exposed to any pattern whatsoever. The square property explains why addictive behaviour remains robust.

This quadratic impact of the degree of learning on the retrieval of the strong pattern (which we have checked to hold also for networks with low average activation) is, in our view, a remarkable property that will play a fundamental role in the new area of research opened up by using strong patterns in an associative memory network to model how behaviour is formed and how it can change including through psychotherapy as explained in reference [15] cited in our paper.